# Tensor Train Decomposition for Adversarial Attacks on Computer Vision Models

## Abstract

Deep neural networks (DNNs) are widely used today, but they are vulnerable to adversarial attacks. To develop effective methods of defense, it is important to understand the potential weak spots of DNNs. Often attacks are organized taking into account the architecture of models (white-box approach) and based on gradient methods, but for real-world DNNs this approach in most cases is impossible. At the same time, several gradient-free optimization algorithms are used to attack black-box models. However, classical methods are often ineffective in the multidimensional case. To organize black-box attacks for computer vision models, in this work, we propose the use of an optimizer based on the low-rank tensor train (TT) format, which has gained popularity in various practical multidimensional applications in recent years. Combined with the attribution of the target image, which is built by the auxiliary (white-box) model, the TT-based optimization method makes it possible to organize an effective black-box attack by small perturbation of pixels in the target image. The superiority of the proposed approach over three popular baselines is demonstrated for seven modern DNNs on the ImageNet dataset.

## 1 Introduction

In recent years, extensive research has been carried out on the topic of adversarial attacks on DNNs, including for classification problems in computer vision as described, for example, in reviews (Qiu et al., 2019; Akhtar et al., 2021; Chakraborty et al., 2021). New successful adversarial attacks allow us to better understand the functioning of DNNs and develop new effective methods of protection against real attacks in various practical applications. White-box attacks, which assume that the internal structure of the attacked DNN is known and the gradient of the target output can be calculated, are very successful in many cases. However, for real-world models, a black-box attack is much more relevant, in which information about the internal structure of the DNN is not used, and it is only possible to make requests to the network and obtain the corresponding outputs.

In the case of black-box attacks, gradient-free optimization methods (Larson et al., 2019) are most often used to find the optimal perturbation of the target image, which leads to a change in the model's prediction. These methods are based either on constructing an estimate of the gradient of the objective function followed by conventional gradient descent, or on various sampling heuristics, including genetic algorithms and evolutionary strategies. However, in the case of a substantially multidimensional search space, these methods may not be effective enough, and it becomes attractive to use the TT-format (Oseledets, 2011), which is a universal approach for manipulating multidimensional data arrays (tensors).

The TT-format provides a wide range of methods for constructing surrogate models and effectively performing linear algebra operations (solution of linear systems, integration, convolution, etc.). Recently it was shown in (Sozykin et al., 2022; Chertkov et al., 2022) that the TT-format can also be successfully applied for optimization problems, and the quality of the solution in many cases turns out to be higher than for methods based on alternative gradient-free approaches. Therefore, in this work, we propose the use of a new TT-based optimization method PROTES (Batsheva et al., 2023) for black-box attacks. To further speed up the computation procedure, we build an attribution map for the target image by a popular Integrated Gradients method (Sundararajan et al., 2017) using an auxiliary pre-trained DNN for which gradients are assumed to be available. In terms of an untargeted attack, we select a set of pixels that have the highest attribution value, and then only change them during the

Figure 1: Schematic representation of the proposed method for black-box adversarial attacks.

optimization process. As a result, we come to a new effective method[1] TETRADAT (**TE**nsor **TR**ain **AD**versarial **AT**tacks) for generating successful adversarial attacks on DNNs (please, see illustration in Figure 1).

This work is organized as follows. In Section 2, we briefly describe the TT-decomposition and gradient-free optimization methods based on it, discuss the attribution problem for DNNs, and then review modern methods of adversarial attacks on computer vision models. Next, in Section 3, we present in detail the proposed TETRADAT method for adversarial attacks on DNN classifiers. In Section 4, we report the results of adversarial attacks performed for seven popular models (including adversarially trained) on the ImageNet dataset with our method and three well-known black-box-based baselines. Finally, in Section 5, we formulate conclusions and discuss further possible improvements and applications of the proposed approach.

## 2 BACKGROUND AND RELATED WORK

In this section, we formulate the problem, describe the main components of the proposed approach (TT-decomposition, TT-based black-box optimization, attribution for DNNs), and then discuss modern methods of adversarial attacks on computer vision models.

We consider adversarial attacks on DNN classifiers in computer vision. An input image of size $d_1 \times d_2$ pixels (for simplicity, we do not take into account color channels) may be represented as a vector[2] $\boldsymbol{x} \in \mathbb{R}^d$, where $d = d_1 \cdot d_2$, and $\boldsymbol{x}[i]$ is a value of the $i$-th pixel ($i = 1, 2, \ldots, d$). When feeding the image $\boldsymbol{x}$ to the DNN, we obtain the probability distribution

$$\boldsymbol{F}(\boldsymbol{x}) \in \mathbb{R}_{[0,1]}^C, \quad \sum_{\tilde{c}=1}^{C} \boldsymbol{F}(\boldsymbol{x})[\tilde{c}] = 1, \tag{1}$$

for $C > 1$ possible classes, where the vector function $\boldsymbol{F}$ denotes the action of the DNN. We define the image class prediction $c$, and the corresponding score $y_c$ as

$$c = \mathsf{argmax}(\boldsymbol{F}(\boldsymbol{x})), \quad y_c = \mathsf{f}(\boldsymbol{x}) \equiv \boldsymbol{F}(\boldsymbol{x})[c] \in \mathbb{R}_{[0,1]}. \tag{2}$$

Then[3] the problem may be formulated as $\boldsymbol{z}_{opt} = \mathsf{min}_{\boldsymbol{z}} \ \mathsf{f}(\boldsymbol{x} + \boldsymbol{z})$, where $\boldsymbol{z} \in \mathbb{R}^d$ is a small perturbation of the target image limited in amplitude, e.g., $||\boldsymbol{z}||_l \leq \epsilon$ ($l = 0, 1, 2, \infty, \epsilon > 0$). The attack is successful if $c \neq \mathsf{argmax}(\boldsymbol{F}(\boldsymbol{x} + \boldsymbol{z}_{opt}))$ for the found optimal perturbation $\boldsymbol{z}_{opt}$.

---

[1]The program code with all numerical examples from this work is publicly available in the repository ANONYMIZED.

[2]We denote vectors with bold letters ($\boldsymbol{a}, \boldsymbol{b}, \boldsymbol{c}, \ldots$), we use upper case letters ($A, B, C, \ldots$) for matrices, and calligraphic upper case letters ($\mathcal{A}, \mathcal{B}, \mathcal{C}, \ldots$) for tensors with $d > 2$. A tensor is just an array with a number of dimensions $d$ ($d \geq 1$); a two-dimensional tensor ($d = 2$) is a matrix, and when $d = 1$ it is a vector. The $(n_1, n_2, \ldots, n_d)$th entry of a $d$-dimensional tensor $\mathcal{Y} \in \mathbb{R}^{N_1 \times N_2 \times \ldots \times N_d}$ is denoted by $\mathcal{Y}[n_1, n_2, \ldots, n_d]$, where $n_k = 1, 2, \ldots, N_k$ ($k = 1, 2, \ldots, d$), and $N_k$ is a size of the $k$-th mode.

[3]In this work, we consider the case of untargeted attacks, and do not deal with targeted attacks, when we need to change the network prediction to a predetermined target class, which is specified by the user.

## 2.1 Tensor Train Decomposition

Let consider the perturbation $z$ as a discrete quantity, i.e.,

$$z[i] \in \left\{ \left( 2 \cdot \frac{n_i - 1}{N_i - 1} - 1 \right) \cdot \epsilon \;\middle|\; n_i = 1, 2, \ldots, N_i \right\} \quad \text{for all } i = 1, 2, \ldots, d, \tag{3}$$

where $N_i$ is a number of possible discrete values for perturbations of the $i$-th pixel, and $\epsilon$ is the maximum amplitude of the perturbation. The values of the function $y = \mathsf{f}(x + z)$ form a tensor $\mathcal{Y} \in \mathbb{R}^{N_1 \times N_2 \times \ldots \times N_d}$ such that $\mathcal{Y}[n_1, n_2, \ldots, n_d] = \mathsf{f}(x + z)$, where $z$ is determined by a multi-index $(n_1, n_2, \ldots, n_d)$ according to (3). Then we come to the problem of discrete optimization, i.e., we search for the minimum of the implicitly specified multidimensional tensor $\mathcal{Y}$, and to solve it we can use the TT-format.

A tensor $\mathcal{Y} \in \mathbb{R}^{N_1 \times N_2 \times \ldots \times N_d}$ is said to be in the TT-format (Oseledets, 2011), if its elements are represented by the following formula

$$\mathcal{Y}[n_1, n_2, \ldots, n_d] = \sum_{r_1=1}^{R_1} \sum_{r_2=1}^{R_2} \cdots \sum_{r_{d-1}=1}^{R_{d-1}} \mathcal{G}_1[1, n_1, r_1] \, \mathcal{G}_2[r_1, n_2, r_2] \, \ldots \mathcal{G}_{d-1}[r_{d-2}, n_{d-1}, r_{d-1}] \, \mathcal{G}_d[r_{d-1}, n_d, 1], \tag{4}$$

where $n = [n_1, n_2, \ldots, n_d]$ is a multi-index ($n_i = 1, 2, \ldots, N_i$ for $i = 1, 2, \ldots, d$), integers $R_0, R_1, \ldots, R_d$ (with convention $R_0 = R_d = 1$) are named TT-ranks, and three-dimensional tensors $\mathcal{G}_i \in \mathbb{R}^{R_{i-1} \times N_i \times R_i}$ ($i = 1, 2, \ldots, d$) are named TT-cores. The TT-decomposition (4) allows to represent a tensor in a compact and descriptive low-parameter form, which is linear in dimension $d$, i.e., it has less than $d \cdot \max_{i=1,\ldots,d}(N_i R_i^2)$ parameters. In addition to reducing memory consumption, the TT-format also makes it possible to obtain a linear in dimension $d$ complexity of many algebra operations, including element-wise addition and multiplication of tensors, calculation of convolution and integration, solving systems of linear equations, calculation of statistical moments, etc.

It is important that there is a set of methods that allow us to quickly construct a TT-decomposition for a tensor specified by a training data set (Chertkov et al., 2023a) or as a function that calculates any of its requested elements (Oseledets & Tyrtyshnikov, 2010). Note that in both cases there is no need to store the full tensor in memory, and the algorithms usually require only a small part of the tensor elements to construct a high-quality approximation. Due to the described properties, the TT-format has become widespread in various practical applications (Cichocki et al., 2017; Chertkov & Oseledets, 2021; Khrulkov et al., 2023), including surrogate modeling, solving differential and integral equations, acceleration and compression of neural networks, etc.

## 2.2 Gradient-free Optimization

TT-format can be applied for black-box optimization problems. In the recent work (Sozykin et al., 2022) the TTOpt algorithm was proposed and its superiority over many popular optimization approaches, including genetic algorithms and evolutionary strategies, was shown. A more powerful algorithm PROTES was presented in later work (Batsheva et al., 2023). The applicability of these methods to various problems was considered in (Nikitin et al., 2022; Chertkov et al., 2022; 2023b; Morozov et al., 2023; Belokonev et al., 2023).

In PROTES, the discrete non-normalized probability distribution $\mathcal{P}_\theta \in \mathbb{R}^{N_1 \times N_2 \times \ldots \times N_d}$ is introduced for the tensor $\mathcal{Y}$ being optimized. The distribution $\mathcal{P}_\theta$ is represented in the TT-format (4), and parameters $\theta$ relate to the values of the corresponding TT-cores. Iterations start from a random non-negative TT-tensor $\mathcal{P}_\theta$ and the following steps are iteratively performed until the budget is exhausted or until convergence (please, see illustration in Figure 2):

1. Sample $K$ candidates of $n_{min}$ from the current distribution $\mathcal{P}_\theta$: $\mathcal{N}^{(K)} = \{n^{(1)}, \ldots, n^{(K)}\}$;

2. Compute the corresponding black-box values $\{y^{(1)}, \ldots, y^{(K)}\}$;

3. Select $k$ best candidates with indices $\mathcal{S} = \{s_1, \ldots, s_k\}$ from $\mathcal{N}^{(K)}$ with the minimal objective value, i.e., $y^{(j)} \leq y^{(J)}$ for all $j \in \mathcal{S}$ and $J \in \{1, \ldots, K\} \setminus \mathcal{S}$;

Figure 2: Schematic representation of the optimization method PROTES.

4. Update the probability distribution $\mathcal{P}_\theta$ ($\theta \to \theta^{(new)}$) to increase the likelihood of selected candidates $\mathcal{S}$, i.e., perform several ($k_{gd}$) gradient ascent steps with the learning rate $\lambda$ for the tensor $\mathcal{P}_\theta$ with the loss function $\widehat{L}_\theta(\{\boldsymbol{n}^{(s_1)}, \ldots, \boldsymbol{n}^{(s_k)}\}) = \sum_{i=1}^{k} \log\left(\mathcal{P}_\theta[\boldsymbol{n}^{(s_i)}]\right)$.

After a sufficient number of iterations, we expect that the distribution $\mathcal{P}_\theta$ will concentrate near the optimum, and among the values requested by the method, there will be multi-indices close to the exact minimum $\boldsymbol{n}_{min}$. It is important to note that all operations described in the above algorithm can be efficiently performed in the TT-format, which ensures the high performance of the method. The values $K$, $k$, $k_{gd}$, $\lambda$ and the number of parameters in $\theta$ (i.e., the average TT-rank $r$ of the TT-decomposition) are the hyperparameters of the PROTES algorithm, and their choice is discussed in detail in (Batsheva et al., 2023).

## 2.3 Attribution of Neural Networks

Attribution plays an important role in interpreting artificial intelligence models (Zhang et al., 2021; Matveev et al., 2021). It allows for a given input image $\boldsymbol{x} \in \mathbb{R}^d$ to construct an attribution map, that is, the vector $\boldsymbol{a} \in \mathbb{R}^d$, such that $\boldsymbol{a}[i]$ is the degree of significance of the $i$-th pixel on the model score $\mathsf{f}(\boldsymbol{x}) \equiv \boldsymbol{F}(\boldsymbol{x})[c]$ for the image class prediction $c$. In the context of adversarial attacks, attribution is of interest because it allows us to select only a part $\hat{d}$ of the pixels of the image based on their semantic significance for the attack.

The Saliency Map method (Simonyan et al., 2013) is a basic approach to white-box model attribution in which the derivative of the DNN output is computed for each pixel of the input image, i.e.,

$$\boldsymbol{a}[i] = \frac{\partial\,\mathsf{f}(\boldsymbol{x})}{\partial\,\boldsymbol{x}[i]}, \quad \textit{for all } i = 1, 2, \ldots, d.$$

However, such a naive approach in some cases turns out to be insufficiently accurate, for example in local extrema and flat regions of the score $\mathsf{f}(\boldsymbol{x})$.

The more successful method is Integrated Gradients (Sundararajan et al., 2017) in which a sequence of linear transformations of the image $\boldsymbol{x}$ into a baseline $\boldsymbol{x}'$ (a completely black image or a noise-based image) is considered. Then the attribution is computed as the averaging of gradients over all transformed images

$$\boldsymbol{a}[i] = (\boldsymbol{x}[i] - \boldsymbol{x}'[i]) \cdot \int_{\xi=0}^{1} \frac{\partial\,\mathsf{f}\left(\boldsymbol{x}' + \xi \cdot (\boldsymbol{x} - \boldsymbol{x}')\right)}{\partial\,\boldsymbol{x}[i]}\, d\xi, \quad \textit{for all } i = 1, 2, \ldots, d. \tag{5}$$

As proven in (Sundararajan et al., 2017), such attribution has useful properties, in particular, high sensitivity and independence from the internal implementation of the DNN. Therefore, we use the Integrated Gradients method as part of our approach for adversarial attacks. Its hyperparameters are the number of gradient descent steps for approximate derivative calculation and the number of discretization nodes in (5).

## 2.4 BLACK-BOX ADVERSARIAL ATTACKS

Black-box attacks can be divided into query-based attacks and transfer-based attacks. White-box auxiliary models and existing white-box attacks are often used for transfer-based attacks to generate adversarial perturbations, which are then fed to the target DNN in a black-box setting. The effectiveness of such attacks depends heavily on transferability across different models, and it seems unlikely that a great deal of knowledge can be extracted from one auxiliary model. As the auxiliary model is trained to be representative of an attacked DNN in terms of its classification rules, it becomes computationally expensive and hardly feasible when attacking large models. To increase the transferability of such attacks, various methods are proposed to escape from poor local optima of white-box optimizer (Dong et al., 2018) or increase the diversity of candidates with augmentation-based methods (Wang & He, 2021).

In the case of query-based attacks, only interaction with the target model is carried out to generate an adversarial example, and various methods for approximate gradient construction followed by gradient descent (Ilyas et al., 2018), or gradient-free heuristics (Su et al., 2019; Andriushchenko et al., 2020; Pomponi et al., 2022) can be used. The heuristic-based approach is usually more query-efficient since estimating the gradient (with finite differences or stochastic coordinate descent) requires additional queries to the DNN.

We classify our method TETRADAT as a query-based method with gradient-free heuristics, despite the use of an auxiliary white-box model, since we need it only to identify the semantic features of the image and its internal structure (provided that the semantic map is correct) has little impact on the quality of our results. Therefore, as the closest alternative approaches we consider Onepixel, Square, and Pixle methods.

The Onepixel attack algorithm (Su et al., 2019) applies the differential evolution (Storn & Price, 1997) to search the suitable pixels and only modify them to perturb target images. The vector norm of perturbation when using this method is rather small, but attacks may be perceptible to people since the changes in the image are very intense and irregular.

The Square attack algorithm (Andriushchenko et al., 2020) performs randomly selected squared perturbations, i.e., it selects localized square-shaped updates of the target image at random positions so that at each iteration the perturbation is situated approximately at the boundary of the feasible set. This method turns out to be query efficient, but the distortion it introduces into the target image is in many cases easily distinguishable visually.

The Pixle attack algorithm (Pomponi et al., 2022) generates adversarial examples by rearranging a small number of pixels using random search, i.e., a patch of pixels is iteratively sampled from the target image, and then some pixels within it are rearranged using a predefined heuristic function. The specificity of image changes in this approach potentially makes it possible to obtain adversarial examples that are practically visually indistinguishable from the original image, however, in some cases this heuristic may not work effectively.

## 3 ADVERSARIAL ATTACKS WITH TENSOR TRAINS

The proposed method TETRADAT for black-box adversarial attacks is schematically illustrated in Figure 1 and presented in Algorithms 1, and 2. First, we establish the top class $c$ predicted by the attacked model $\boldsymbol{F}$ for the given input image $\boldsymbol{x}$ having $d$ pixels. Then we build an attribution map $\boldsymbol{a}$ using the auxiliary DNN (white-box) $\hat{\boldsymbol{F}}$ by the Integrated Gradients method [4] and select $\hat{d}$ ($\hat{d} \leq d$) pixel positions $\boldsymbol{q}$ with the highest attribution value.

Next, we carry out successive runs of the PROTES optimizer, each time reducing the amplitude of the perturbation $\epsilon$ by half until the budget $m$ is exhausted. Each run of the optimizer continues until a perturbation $\boldsymbol{n}_{adv}$ appears, which leads to a successful attack. During restarts, we initialize the PROTES optimizer by the probabilistic tensor in the TT-format $\mathcal{P}_\theta$ obtained at the end of the previous

---

[4] It should be especially noted that in this case, gradients are calculated not for the attacked (black-box) model, but for an auxiliary (white-box) model, which can be significantly simpler than the attacked model. The choice of the auxiliary model has little impact on the attack success rate provided that it correctly recognizes the image class, and therefore the attribution leads to a relevant semantic map.

---

**Algorithm 1:** The TETRADAT method for black-box adversarial attacks.

---

**Data:** attacked DNN $\boldsymbol{F}$; auxiliary DNN for attribution $\hat{\boldsymbol{F}}$; input image $\boldsymbol{x}$ having $d$ pixels; maximum number of perturbed pixels $\hat{d}$; initial amplitude of the perturbation $\epsilon$; maximum number of queries $m$ (i.e., the computational budget).

**Result:** adversarial image $\boldsymbol{x}_{adv}$.

1 Find the image class prediction: $c = \mathsf{argmax}\left(\boldsymbol{F}(\boldsymbol{x})\right)$

2 Build the attribution map for the class $c$: $\boldsymbol{a} = \mathsf{integrated\_gradients}\left(\hat{\boldsymbol{F}}, \boldsymbol{x}, c\right)$

3 Select $\hat{d}$ pixels with the highest attribution value: $\boldsymbol{q} = \mathsf{argsort}_{descending}(\boldsymbol{a})[1:\hat{d}]$

4 Generate a random non-negative TT-tensor: $\mathcal{P}_\theta \in \mathbb{R}^{N_1 \times N_2 \times \ldots \times N_{\hat{d}}}$ ($N_1 = \ldots = N_{\hat{d}} = 3$)

5 **while** *the budget $m$ is not exhausted* **do**

6    **Function** `loss(n)`:

7       Apply the perturbation: $\boldsymbol{x}_{new} = \mathsf{perturb}(\boldsymbol{x}, \boldsymbol{n}, \boldsymbol{q}, \epsilon)$ // See Algorithm 2

8       Compute the new score for the class $c$: $y = \boldsymbol{F}(\boldsymbol{x}_{new})[c]$

9       **return** $y$

10    Optimize until the attack is successful: $\boldsymbol{n}_{adv}, \mathcal{P}_\theta = \mathsf{protes}(\mathsf{loss}, \mathcal{P}_\theta)$

11    Decrease the amplitude of the perturbation: $\epsilon = \epsilon/2$

12 **end**

13 Apply the found optimal perturbation: $\boldsymbol{x}_{adv} = \mathsf{perturb}(\boldsymbol{x}, \boldsymbol{n}_{adv}, \boldsymbol{q}, \epsilon)$ // See Algorithm 2

---

---

**Algorithm 2:** Image perturbation in the TETRADAT method.

---

**Data:** input image $\boldsymbol{x}$ having $d$ pixels; the multi-index of discrete perturbation $\boldsymbol{n} \in \mathbb{R}^{\hat{d}}$; list of used $\hat{d}$ pixel numbers $\boldsymbol{q}$; amplitude of the perturbation $\epsilon$.

**Result:** the perturbed image $\boldsymbol{x}_{new}$.

1 Copy the input image: $\boldsymbol{x}_{new} \leftarrow \boldsymbol{x}$

2 **for** $i = 1$ **to** $\hat{d}$ **do**

3    Compute HSV representation of the pixel: $H, S, V = \mathsf{rgb\_to\_hsv}(\boldsymbol{x}[\boldsymbol{q}[i]])$

4    If $\boldsymbol{n}[i] = 1$, then decrease the S-channel intensity: $S \leftarrow \max\left(0, S - \epsilon\right)$

5    If $\boldsymbol{n}[i] = 2$, then do not change the intensity

6    If $\boldsymbol{n}[i] = 3$, then increase the V-channel intensity: $V \leftarrow \min\left(1, V + \epsilon\right)$

7    Update pixel intensity from HSV representation: $\boldsymbol{x}_{new}[\boldsymbol{q}[i]] = \mathsf{hsv\_to\_rgb}(H, S, V)$

8 **end**

---

step (on the first run we generate a random TT-tensor). This strategy allows us to effectively reuse information on each subsequent run, and achieve a balance between the attack success rate and the smallness of the perturbations.

With PROTES, we minimize the function $\mathsf{loss}$, which returns the score of the attacked DNN for the proposed discrete perturbation multi-index[5] $\boldsymbol{n} \in \mathbb{R}^{\hat{d}}$, as presented in Algorithm 1. We consider the simplest case of a discrete grid like (3) with 3 nodes, that is, for each pixel there are three options: decrease the value by $\epsilon$ (for the index 1), do not change the value (for the index 2), and increase the value by $\epsilon$ (for the index 3). We can perturb the pixel values in the same way for all three RGB (Red, Green, Blue) color channels at once or consider the optimization of each color channel separately, but the following approach turns out to be more useful in terms of attack success rate and visual assessment of generated adversarial images. As presented in Algorithm 2, we convert the RGB value to the HSV (Hue, Saturation, Value) color model. Then for the case of decreasing intensity we change the S-factor, and for the case of increasing intensity, we change the V-factor.

Thus, the hyperparameters of our method are the DNN $\hat{\boldsymbol{F}}$ used for attribution (as mentioned above, the result is practically independent of the choice of model, provided that it builds a semantically correct attribution map), the initial amplitude of the perturbation $\epsilon$ (for simplicity, we always take it equal to 1), the number of pixels used for optimization $\hat{d}$ (empirically it turns out that 10 percent of the total number of pixels is enough), as well as hyperparameters of the Integrated Gradients and PROTES method.

---

[5]PROTES requests a batch of values at once, but for simplicity, we demonstrate only one multi-index.

Table 1: Attack success rate for the baselines and the proposed TETRADAT method.

|  | Onepixel | Pixle | Square | TETRADAT |
| --- | --- | --- | --- | --- |
| Alexnet | 27.60 % | 100.00 % | 94.22 % | 99.28 % |
| Googlenet | 26.95 % | 98.41 % | 96.30 % | 99.08 % |
| Inception | 30.12 % | 94.11 % | 91.97 % | 97.86 % |
| Mobilenet | 9.99 % | 92.16 % | 96.08 % | 97.22 % |
| Resnet | 4.69 % | 61.49 % | 80.26 % | 85.68 % |
| Adv. Inception | 43.17 % | 94.82 % | 92.66 % | 98.56 % |
| Adv. Inception-Resnet | 32.06 % | 84.89 % | 78.98 % | 95.80 % |

Table 2: $L_1$ norm of the perturbations averaged over all successful attacks for the baselines and the proposed TETRADAT method.

|  | Onepixel | Pixle | Square | TETRADAT |
| --- | --- | --- | --- | --- |
| Alexnet | 449.6 | 1961.2 | 10251.9 | 2222.0 |
| Googlenet | 441.8 | 1705.2 | 10246.8 | 1750.9 |
| Inception | 434.4 | 1660.4 | 10242.0 | 1834.0 |
| Mobilenet | 433.8 | 3051.3 | 10249.6 | 1890.8 |
| Resnet | 435.3 | 3061.9 | 10249.2 | 3495.0 |
| Adv. Inception | 432.3 | 1557.2 | 10270.2 | 2019.8 |
| Adv. Inception-Resnet | 431.2 | 1941.5 | 10278.2 | 2680.7 |

Table 3: $L_2$ norm of the perturbations averaged over all successful attacks for the baselines and the proposed TETRADAT method.

|  | Onepixel | Pixle | Square | TETRADAT |
| --- | --- | --- | --- | --- |
| Alexnet | 31.4 | 57.2 | 26.7 | 33.1 |
| Googlenet | 31.0 | 53.1 | 26.6 | 26.3 |
| Inception | 30.5 | 50.8 | 26.6 | 27.4 |
| Mobilenet | 30.4 | 70.8 | 26.6 | 28.2 |
| Resnet | 30.3 | 72.4 | 26.6 | 51.7 |
| Adv. Inception | 30.4 | 49.1 | 26.7 | 30.1 |
| Adv. Inception-Resnet | 30.3 | 55.2 | 26.7 | 40.0 |

## 4 NUMERICAL EXPERIMENTS

To evaluate the performance of the proposed method TETRADAT, we conducted a series of numerical experiments in the following setting.

**Dataset.** We consider the ImageNet dataset (Deng et al., 2009) since it corresponds to high-resolution images of real objects, attacks on which seem relevant in practical applications.

**Models for attack.** We consider five classic popular architectures[6]: Alexnet (Krizhevsky, 2014), Googlenet (Szegedy et al., 2015), Inception V3 (Szegedy et al., 2016), Mobilenet V3 LARGE (Howard et al., 2019), and Resnet 152 (He et al., 2016), as well as two adversarially trained models: Adversarial Inception[7] and Adversarial Inception ResNet[8], for validation of our method.

**Model for attribution.** We consider the pre-trained VGG 19 (Simonyan & Zisserman, 2014) model from the torchvision package to build an attribution map. Note that the results do not depend

---

[6]In experiments, we used pre-trained models from the torchvision package `https://github.com/pytorch/vision`.

[7]See `https://huggingface.co/docs/timm/en/models/adversarial-inception-v3`.

[8]See `https://huggingface.co/docs/timm/en/models/ensemble-adversarial`.

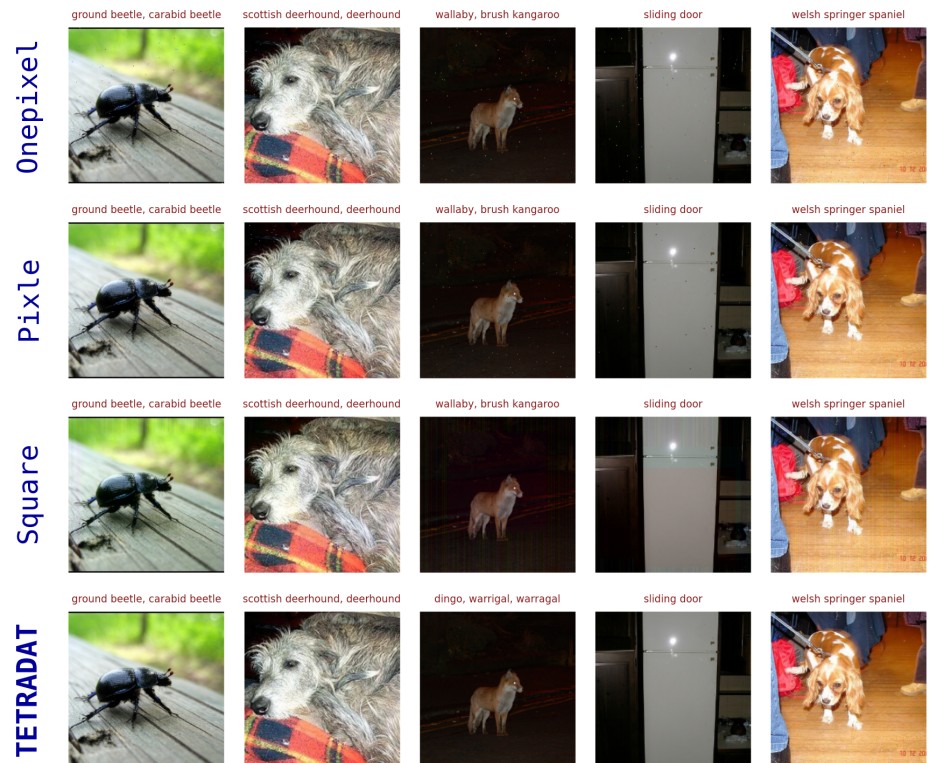

Figure 3: Visualization of results for five random successful adversarial attacks on the Mobilenet. The correct classes for the presented images correspond to (from left to right): "dung beetle", "irish wolfhound", "red fox", "refrigerator, icebox", "blenheim spaniel". The network's predictions for the attacked images are presented in the corresponding titles.

significantly on the choice of model, provided that it allows constructing an attribution map that is correct from a semantic point of view.

**Images for attack.** We consider one image from each of the 1000 classes in the ImageNet dataset,[9] and the attack was carried out only if the corresponding class was correctly predicted by the attacked model and by the model for attribution[10]. Thus, for Alexnet we used 692 images for attack, for Googlenet – 757 images, for Inception – 747 images, for Mobilenet – 791 images, and for Resnet – 831 images, for Adversarial Inception – 695 images, and for Adversarial Inception ResNet – 761 images. The top-1 accuracy on the used set of images is 74.2% for Alexnet, 79.5% for Googlenet, 82.0% for Inception, 84.7% for Mobilenet, 95.0% for Resnet, 76.1% for Adversarial Inception, 83.3% for Adversarial Inception ResNet, and 84.0% for VGG.

**Hyperparameters of the Integrated Gradients method.** We used our implementation of the method, with 15 gradient descent steps, and 15 nodes for discretization. For subsequent optimization, we selected $\hat{d} = 5000$ pixels with the highest attribution value (that is, approximately 10% of the number of pixels in the image).

**Hyperparameters of the PROTES method.** We used the official implementation of the method[11] with the following parameters for all computations: $K = 100$, $k = 10$, $k_{gd} = 100$, $\lambda = 0.01$,

---

[9]The images from repository https://github.com/EliSchwartz/imagenet-sample-images.git are used.

[10]Excluding from consideration images in which the auxiliary model gives an incorrect prediction does not limit the generality of the method, since in this case, we can use any other model that gives the correct prediction for the image or even manually select semantic areas of significance in the image.

[11]See repository https://github.com/anabatsh/PROTES.

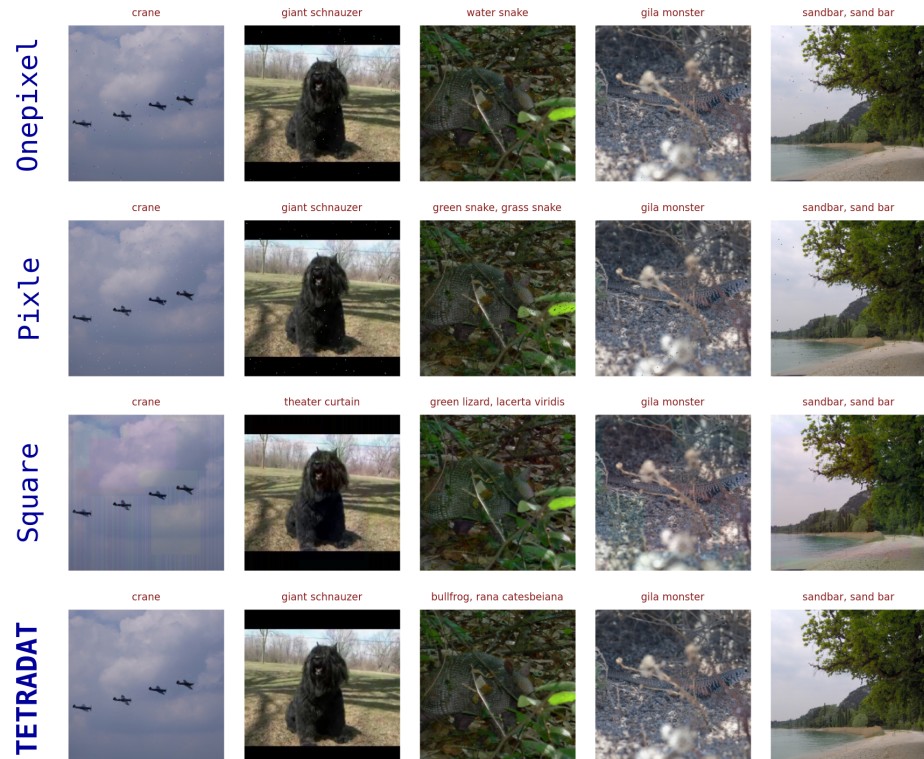

Figure 4: Visualization of results for five random successful adversarial attacks on the Resnet. The correct classes for the presented images correspond to (from left to right): "warplane, military plane", "bouvier des flandres", "armadillo", "whiptail", "lakeside". The network's predictions for the attacked images are presented in the corresponding titles.

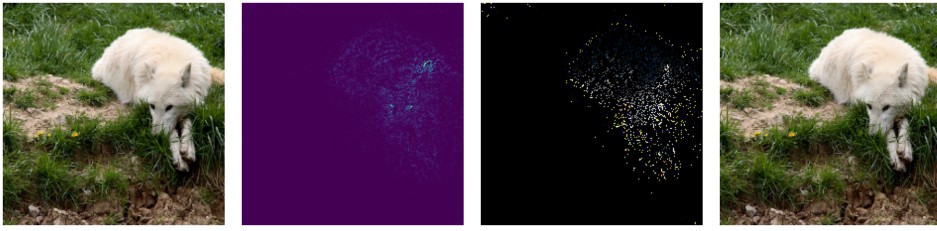

Figure 5: The result of a randomly selected successful attack on the Mobilenet with TETRADAT. We present (from left to right): the original image (the correct class is "white wolf, arctic wolf, canis lupus tundrarum"), the computed attribution map from the auxiliary model, the proposed image perturbation (pixel intensity is increased 10 times for visibility), and the final adversarial image (the related prediction of the model is "timber wolf, grey wolf, gray wolf, canis lupus").

and $r = 5$ (please see the description of these parameters in the section dedicated to the PROTES optimizer above). The computational budget for the PROTES method (that is, the number of requests to the black-box model) was limited to $10^4$.

**Baselines.** We consider three well-known black-box methods: Onepixel, Pixle, and Square, which were discussed in the literature review section above. We used implementations from a popular library torchattacks (Kim, 2020), and set the default hyperparameters for all methods, with the following exceptions. For the Onepixel method, we increased the number of attacked pixels from 1 to 100, since with a lower value the percentage of successful attacks is too low, and with a higher value the distortion of the original image becomes too large. For the Square method, we reduced the maximum

perturbation value from $8/255$ to $4/255$, since otherwise the distortions of the attacked image turn out to be dramatic (bright stripes and rectangles running throughout the entire image). The computational budget for all methods was limited to $10^4$.

**Results.** We report the attack success rates for all considered DNNs and methods in Table 1. The average $L_1$ and $L_2$ norms of the perturbations are presented in Table 2, and Table 3 respectively. Generated adversarial images for five random successful adversarial attacks for the Mobilenet and Resnet model are visualized in Figure 3, and Figure 4 respectively. In Figure 5 we demonstrate an example of the attack on the Mobilenet model with our method.

**Discussion.** As follows from the presented results, the proposed method gives a higher percentage of successful attacks in six out of seven cases (for the Alexnet model, the Pixle method was slightly more accurate). According to the $L_1$ norm, our method is significantly superior to the Square method, and according to the $L_2$ norm, it is better than the Pixle method, while visually TETRADAT produces the most realistic adversarial images (this becomes more noticeable as images are zoomed in).

## 5 CONCLUSIONS

In this work, we presented a new method TETRADAT for untargeted adversarial query-based black-box attacks on computer vision models. Our approach is based on gradient-free optimization with the low-rank tensor train format and additional dimensionality reduction by using the semantic attribution map of the target image. TETRADAT with a constant set of hyperparameters demonstrates a high percentage of successful attacks for various modern neural network models in comparison with well-known alternative black-box methods, while image distortion in many cases turns out to be negligible. As part of the further work, we plan to extend our approach to targeted attacks and other relevant problems, including label-based and universal attacks.

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
