# OpenReview forum: "Tensor Train Decomposition for Adversarial Attacks on Computer Vision Models"
_ICLR.cc/2025/Conference — Submitted to ICLR 2025_

### Official Review · Reviewer_Jga9 · 2024-10-29

**Soundness:** 1
**Presentation:** 2
**Contribution:** 1
**Rating:** 3
**Confidence:** 5

**Summary:**

This paper introduces TETRADAT, a novel method for conducting black-box adversarial attacks on deep neural networks (DNNs) in the field of computer vision. The method leverages the Tensor Train (TT) decomposition, a format popular for its efficiency in handling multidimensional data, combined with an attribution map generated by an auxiliary white-box model. The authors propose using the PROTES optimizer, a gradient-free technique, to perturb pixels in the target image that have the highest attribution value, thereby causing misclassification by the DNN with minimal distortion.

**Strengths:**

1. The paper is easy to understand.

The paper is logically expressed, and the description of the method is complete and intuitive. The authors explain the principles and details of the TETRADAT with pictures and text, which makes it easy to understand.

**Weaknesses:**

1. The method proposed in this paper is not innovative enough.

(1) While the paper introduces the TETRADAT method for performing black-box adversarial attacks, it relies primarily on the PROTES optimiser, as described in the paper by Batsheva et al.[1], without making any innovative modifications to the PROTES algorithm itself.

(2) While this paper presents an approach that integrates a saliency map with the PROTES optimisation framework to perform adversarial attacks on deep neural networks, the concept of using saliency maps for such attacks is not new. Previous works, such as Liu et al.[2] and Dai et al.[3], have explored the use of saliency maps in the context of adversarial attacks.

2. There are some issues with the experimental section of this paper.

(1) Insufficient Benchmarking:
The paper utilizes Onepixel, Pixle, and Square as baselines for comparison, all of which focus on perturbing a small number of pixels to achieve adversarial attacks. However, the paper lacks a comparison with query-based black-box attack methods, such as the approach presented by Bai et al. [4].

(2) Absence of Ablation Studies:
The paper combines the Tensor Train-based methods with the attribution maps generated by auxiliary white-box models. However, there is a noticeable absence of ablation studies that demonstrate the effectiveness of incorporating the attribution maps. Ablation studies are crucial for understanding the incremental benefits of the components included in the proposed approach.

(3) Lack of Result Analysis:
The paper presents results in Table 1, Table 2, Table 3, Figures 3, and Figure 4, but there is a lack of textual analysis to accompany these results. Moreover, Figures 3 and 4 seem to convey the same information, which may indicate a redundancy in the presentation of results. A thorough discussion of the results, including the implications and insights gained from the experimental data, is missing.

(4) Poor Experimental Results:
The average L1 and L2 norms reported for the TETRADAT method in Tables 2 and 3 do not appear to demonstrate a clear advantage over the baseline methods. A discussion on why the TETRADAT method did not outperform the baselines in terms of these norms is necessary to assess the practical applicability of the proposed technique.

3. Some of the views are not supported by evidence.

The paper claims that TETRADAT produces the most realistic adversarial images (this becomes more noticeable as images are zoomed in), as stated in lines 496-500. However, this claim lacks evidence, as the authors have not provided any experiments with human observers to back up the assertion that TETRADAT's images are more realistically altered.

**References**

[1] Batsheva A, Chertkov A, Ryzhakov G, et al. PROTES: probabilistic optimization with tensor sampling[J]. Advances in Neural Information Processing Systems, 2023, 36: 808-823.

[2] Liu H, Zuo X, Huang H, et al. Saliency Map-Based Local White-Box Adversarial Attack Against Deep Neural Networks[C]//CAAI International Conference on Artificial Intelligence. Cham: Springer Nature Switzerland, 2022: 3-14.

[3] Dai Z, Liu S, Li Q, et al. Saliency attack: towards imperceptible black-box adversarial attack[J]. ACM Transactions on Intelligent Systems and Technology, 2023, 14(3): 1-20.

[4] Bai Y, Wang Y, Zeng Y, et al. Query efficient black-box adversarial attack on deep neural networks[J]. Pattern Recognition, 2023, 133: 109037.

**Questions:**

1. Could the authors elaborate on the novel aspects of TETRADAT that differentiate it from the original PROTES framework or other existing works?


2. Is this method superior to advanced query-based black box attack methods?

3. The paper describes the integration of attribution maps with the PROTES optimizer but does not include any ablation studies. Could the authors explain the rationale behind this omission?

4. Regarding the results presented, it seems that Figures 3 and 4 convey similar information. Could the authors clarify the distinction between these two figures and justify their separate inclusion?

5. The L1 and L2 norm results do not clearly demonstrate the superiority of the TETRADAT method over the baselines. Could the authors offer an explanation for this observation and discuss the implications for the practical application of their method?

6. Could the authors elaborate on the basis for their claim that TETRADAT produces the most realistic adversarial images? What observations or preliminary data led to this conclusion?

---

### Official Review · Reviewer_2AFt · 2024-11-02

**Soundness:** 3
**Presentation:** 2
**Contribution:** 2
**Rating:** 5
**Confidence:** 4

**Summary:**

This paper presents a novel method for using black-box adversarial attacks on the deep neural networks. Unlike the traditional methds that rely on the gradient information or gradient-free optimization methods, this paper uses Tensor Train decomposition for high-dimensional data. It is able to attack the network even without fully access the model. By using low-rangk optimizer and the surrogate model, this paper can generate high-impact pertubations and targeted attack the network with efficiency.

**Strengths:**

This method outperforms three baseline methods (Onepixel, Square, and Pixle) in attack tasks across seven models, including adversarially trained models.

This method focuses only on the key pixels, keeping the adversarial attacks not being observed by human eyes, which allows it to achieve high success rate with minimal image distortion.

**Weaknesses:**

The weakness of the paper is that the study focuses on untargeted attacks. It also needs to consider and compare with the targeted attack results. In adddition, the paper does not address if the approach is feasible for larger and deeper models.

**Questions:**

1. This method claimed the high attack success rate. Could the author further clarify the soecific advantages of TT decomposition over other decomposition methods? What's the main different/novel points with the TT decomposition method?
	2. For creating the attribution maps, would the accuracy significantly different if use a different model? For example, can we use transformer-based model, ViT-B/L?
	3. Because this paper is focusing on untargeted attacks, is this method also works with the target attack? Because the target attack is more challenge and it is able to showcase the achievement of the purposed method.
	4. Do the authors have insights on applying this method on larger/deeper model? Such as ViT-L or different domains model like NLP?
	5. This method is very relying on the auxiliary DNN model. If using multiple auxiliary DNN models, will it improve the attack accuracy?
As for generating the perturbating examples, how long it needs to take for generate one sample? If increase the iteration steps, what's the accuracy difference?

---

### Official Review · Reviewer_GgW5 · 2024-11-03

**Soundness:** 2
**Presentation:** 1
**Contribution:** 2
**Rating:** 3
**Confidence:** 5

**Summary:**

The paper presents TETRADAT, a novel black-box adversarial attack method for Deep Neural Networks (DNNs) that doesn’t require model details. Using Tensor Train (TT) decomposition and an optimization technique called PROTES, TETRADAT efficiently crafts small perturbations to fool DNNs with limited queries. It outperforms existing black-box methods in terms of success rates and computational efficiency across various models on the ImageNet dataset. TETRADAT is effective for testing DNN vulnerabilities, offering improved robustness evaluation for AI models.

**Strengths:**

- The paper introduces a unique combination of Tensor Train (TT) decomposition and PROTES optimization to handle high-dimensional perturbations efficiently in black-box settings. According to the experimental results, the proposed method performs better than some query-based methods.

**Weaknesses:**

- The authors classify TETRADAT as a query-based black-box attack method due to its reliance on querying the target model during the optimization process. However, since TETRADAT also uses an auxiliary white-box model to guide the attack, it shares characteristics with transfer-based methods, which typically require no queries to the target model.
Given this similarity, a fairer and more comprehensive evaluation would involve comparing TETRADAT with transfer-based attack methods. This would provide insights into whether querying the target model yields significant advantages over transfer-based approaches, especially when both methods assume access to a white-box auxiliary model.

- While the paper fixes a query budget of $10^4$ for TETRADAT and other baseline methods, it does not provide a detailed comparison of the actual number of queries required to achieve a successful attack across different methods.
Query efficiency is a critical measure of black-box attack performance, especially in scenarios where query access is limited or costly. Without a direct comparison of query counts, it is difficult to assess whether TETRADAT offers an advantage in terms of query efficiency relative to other black-box methods.

- TETRADAT relies on several hyperparameters, including TT ranks, perturbation amplitude, learning rate for the PROTES optimization, and the number of discrete perturbation levels. These hyperparameters require careful tuning to achieve optimal performance.
The complexity of tuning these parameters makes the method less practical for deployment, especially for users without extensive experience in hyperparameter optimization. Additionally, there is limited analysis of how these hyperparameters affect the attack's effectiveness, which could have provided valuable guidance for setting these values.

- TETRADAT discretizes the perturbation levels for each pixel, simplifying the search space but potentially limiting the granularity of the perturbations. The chosen number of discrete levels influences both the subtlety and effectiveness of the attack.
This discretization could restrict the method's flexibility to find the optimal perturbation, particularly in cases where finer adjustments are needed. Continuous optimization methods might perform better in scenarios requiring precise control over perturbation values.

- The evaluation is limited to ImageNet, lacking tests on additional datasets, which would demonstrate the method’s generalizability. A more diverse evaluation, including tests with varied hyperparameter settings, would offer a fuller understanding of TETRADAT’s robustness and effectiveness. Besides, a more detailed analysis of experimental results would be preferred.

- The presentation of this paper doesn't meet the ICLR standards and needs further improvement.

**Questions:**

- How is the auxiliary model selected or trained? Is it a model with the same architecture as the target, or is it a simpler/standard model trained on the same data?
- What are the parameters of the PROTES optimization process? For example, the learning rate, or convergence criteria. Understanding these would clarify the practical tuning required to implement the attack.
- Would using alternative tensor decomposition techniques (e.g., CP or Tucker decomposition) impact performance? It would be interesting to know if the authors tested other decomposition methods or if TT was chosen based on specific benefits in this context.
- How effective is TETRADAT against adversarial defenses? Did the authors test the method against defenses like adversarial training or detection algorithms, and if so, how resilient is it?

**Details Of Ethics Concerns:**

TETRADAT enables subtle black-box attacks that could potentially mislead AI systems in applications like security or diagnostics.

---

### Official Review · Reviewer_DdFF · 2024-11-04

**Soundness:** 2
**Presentation:** 2
**Contribution:** 2
**Rating:** 3
**Confidence:** 4

**Summary:**

This paper proposes a gradient-free adversarial attack method. Their approach is based on the low-rank tensor train format and combined with the attribution of the target image with any auxiliary model. Their approach can devise an effective black-box attack by small perturbation of pixels in the target image. Compared with three popular baselines on the ImageNet dataset.

**Strengths:**

Gradient-free adversarial attack is interesting.

**Weaknesses:**

The experiment scope is too narrow.

Motivation is not clear.

Time complexity is missing.

**Questions:**

1 The experiment scope is too narrow. You consider using IG to generate attribution. Thus, you introduce extra information compared with the baselines in your experiment.Additionally, you should consider comparing with transfer-based attacks with VGG as the source model. Especially, you should consider to compare with attribution-based adversarial attacks [1, 2]. Otherwise, the effectiveness of your approach cannot be fully validated. Furthermore, the victim models are outdated, so you should also consider transformer-based models to attack.

2 What is the benefit of your approach? You adapt TT to the field of adversarial attacks, but you fail to demonstrate the benefit of the adoption of TT-based transformation. You should further emphasize your motivation for this paper.

3 What is the time complexity of your approach? It turns out that your approach needs black-box optimization and searching. Thus, you should illustrate the time efficiency compared with baselines.

4 Why do you keep the hyper-parameter setting of the PROTES instead of searching for the optimal setting under the adversarial attack scenario? If the hyper-parameters are already optimal, you should show ablation studies on those hyper-parameters.

5 The format is a little bit weird. The template of equations seems to be done by Word instead of Latex.

[1] Transferable adversarial attack based on integrated gradients. ICLR 2022.

[2] Improving adversarial transferability via neuron attribution-based attacks. CVPR 2022.

---

### Meta-Review · Area_Chair_jZBk · 2024-12-20

**Metareview:**

This paper proposes TETRADAT, a gradient-free adversarial attack method based on the low-rank tensor train decomposition, a format popular for its efficiency in handling multidimensional data, combined with an attribution map generated by an auxiliary white-box model. The reviewers think the paper's novelty is limited, and the experimental results are inadequate. For example, since the method uses an auxiliary white-box model to guide the attack, it should also be compared to transfer-based attacks. The paper uses a fixed query budget of $10^4$
 for TETRADAT and other baseline methods, but it does not provide a detailed comparison of the actual number of queries required to achieve a successful attack across different methods. The proposed method also relies on several hyperparameters. Based on the reviewers' comments, I recommend rejection.

**Additional Comments On Reviewer Discussion:**

The authors did not provide a rebuttal to address the reviewers' concerns and questions.

---

### Decision · Program_Chairs · 2025-01-22

Reject